# Confusion2Vec 2.0: Enriching ambiguous spoken language representations with subwords

**Prashanth Gurunath Shivakumar** ⓘ *, **Panayiotis Georgiou** ⓘ, **Shrikanth Narayanan** ⓘ

Department of Electrical and Computer Engineering, University of Southern California, Los Angeles, California, United States of America

* pgurunat@usc.edu

**Data Availability Statement:** The Fisher English Training Speech corpora and the Airline Travel Information Systems benchmark dataset are available from the Linguistic Data Consortium.

## Abstract

Word vector representations enable machines to encode human language for spoken language understanding and processing. Confusion2vec, motivated from human speech production and perception, is a word vector representation which encodes ambiguities present in human spoken language in addition to semantics and syntactic information. Confusion2-vec provides a robust spoken language representation by considering inherent human language ambiguities. In this paper, we propose a novel word vector space estimation by unsupervised learning on lattices output by an automatic speech recognition (ASR) system. We encode each word in Confusion2vec vector space by its constituent subword character n-grams. We show that the subword encoding helps better represent the acoustic perceptual ambiguities in human spoken language via information modeled on lattice-structured ASR output. The usefulness of the proposed Confusion2vec representation is evaluated using analogy and word similarity tasks designed for assessing semantic, syntactic and acoustic word relations. We also show the benefits of subword modeling for acoustic ambiguity representation on the task of spoken language intent detection. The results significantly outperform existing word vector representations when evaluated on erroneous ASR outputs, providing improvements up-to 13.12% relative to previous state-of-the-art in intent detection on ATIS benchmark dataset. We demonstrate that Confusion2vec subword modeling eliminates the need for retraining/adapting the natural language understanding models on ASR transcripts.

## Introduction

Speech is the primary and most natural mode of communication for humans. This makes its use also attractive for human-computer interaction, which in turn requires decoding human language to enable spoken language understanding. Human language is a complex construct involving multiple dimensions of information including semantics, syntax and often contain ambiguities which make it challenging for machine inference of communication intent,

Fisher English Training Speech Part 1 Speech: https://catalog.ldc.upenn.edu/LDC2004S13 Fisher English Training Speech Part 1 Transcripts: https://catalog.ldc.upenn.edu/LDC2004T19 Fisher English Training Part 2, Speech: https://catalog.ldc.upenn.edu/LDC2005S13 Fisher English Training Part 2, Transcripts: https://catalog.ldc.upenn.edu/LDC2005T19 ATIS2: https://catalog.ldc.upenn.edu/LDC93S5 ATIS3 Training Data: https://catalog.ldc.upenn.edu/LDC94S19 ATIS3 Test Data: https://catalog.ldc.upenn.edu/LDC95S26 The evaluation datasets for analogy and similarity tasks are made available under GitHub repository: https://github.com/pgurunath/confusion2vec_2.0.

**Funding:** This work was supported by Simons Foundation under grant 627148 awarded to SN. The funders had no role in study design, data collection and analysis, decision to publish, or preparation of the manuscript.

**Competing interests:** The authors have declared that no competing interests exist.

emotions etc. Several word vector representations have been proposed for effectively describing the human language in the natural language processing community.

Contextual modeling techniques like language modeling, i.e., predicting the next word in the sentence given a window of preceding context, have been shown to model meaningful word representations [1, 2]. Bag-of-word based contextual modeling, where the current word is predicted given both its left and right (local) contexts has shown to capture language semantics and syntax [3]. Similarly, predicting local context from the current word, referred to as skip-gram modeling, is shown to better represent semantic and syntactic distances between words [4]. In [5] log bi-linear models combining global word co-occurrence information and local context information, termed as global vectors (GloVe), is shown to produce meaningful structured vector space. Bi-directional language models are proposed in [6], where internal states of deep neural networks are combined to model complex characteristics of word use and its variance over linguistic contexts. The advantages of bi-directional modeling are further exploited along with self-attention using transformer networks [7] to estimate a representation, termed as BERT (Bidirectional Encoder Representations from Transformers), that has shown its utility on a multitude of natural language understanding tasks [8]. Models such as BERT, ELMo estimate word representations that vary depending on the context, whereas the context-free representations including GloVe and Word2Vec generate a single representation irrespective of the context.

However, most of the word vector representations infer the knowledge through contextual modeling and many of the inherent ambiguities present in human language are often unrecognized or ignored. For instance, from the perspective of spoken language, the ambiguities can be associated with how similar the words sound, i.e., for example, the words "see" and "sea" sound acoustically identical but have different meanings. The ambiguities can also be associated with the underlying speech signal itself due to wide range of acoustic environments involving noise, overlapped speech and channel, room characteristics. These ambiguities often project themselves as errors through ASR systems. Most of the existing word vector representations such as word2vec [3, 4], fasttext [9], GloVe [5], BERT [8], ELMo [6] do not account for the ambiguities present in speech signals and thus degrade while processing the output of noisy ASR transcripts.

Confusion2vec was recently proposed to handle representation ambiguity information present in human language [10]. Confusion2vec is estimated by unsupervised skip-gram training on the ASR output lattices and confusion networks. The analysis of inherent acoustic ambiguity information of the embeddings displayed meaningful interactions between the semantic-syntactic subspace and acoustic similarity subspaces. In [11], the usefulness of the Confusion2vec was confirmed on the task of spoken language intent detection. The Confusion2vec representation significantly outperformed typical word embeddings including word2vec and GloVe when evaluated on noisy ASR transcripts by reducing the classification error rate by approximately 20% relative.

Prior attempts at leveraging information present in word lattices and word confusion networks have been successful for multiple tasks [12–17]. However, they have some limitations, as these prior works estimate the embedding in a supervised manner specifically trained with task specific labels. Consequently, the main downside is that the word representation estimated by such techniques are task-dependent and are restricted to a particular domain and dataset. Moreover, availability of most of the task specific datasets are limited and task specific speech data are expensive to collect. The advantage of Confusion2Vec is that it estimates a generic, task-independent word vector representation via unsupervised learning on lattices or confusion networks generated by an ASR on any speech conversations.

In this paper, we extend the previously proposed Confusion2Vec representation framework by incorporating subwords to represent each word for modeling both the acoustic ambiguity information and the contextual information. Each word is modeled as a sum of constituent n-gram characters. Throughout this paper we refer to character n-grams as subwords. Our motivation behind the use of subwords are the following: (i) they incorporate morphological information of the words by encoding internal structure of words [9], (ii) the acoustically ambiguous words tend to have more similar bag-of-character n-grams, (iii) subwords help model under-represented words more efficiently, thereby leading to more robust estimation with limited available data [9], which is the case since training Confusion2Vec is restricted to ASR lattice outputs, (iv) subwords enable representations for out-of-vocabulary words [18] which are commonplace with end-to-end ASR systems outputting characters.

Although the use of subword (character n-grams) may appear commonplace in the NLP domain in terms of practicality, to the best of our knowledge, our work is the first to explore encoding n-gram characters to model acoustic ambiguity jointly with natural language semantics. Unlike typical applications of distributed semantics, in our case, the subword is mapped to a much larger distribution of words, and in some cases observe conflicting semantics and acoustic interactions which render the modeling task more complex and challenging. From an alternative perspective, we model information projected from multiple modalities, i.e., acoustics and natural language, which embeds inherent information including audio channel characteristics, room impulse response, noise environments etc., which further adds to the modeling richness and novelty of the proposed work.

The rest of the paper is organized as follows: Confusion2vec is introduced in Section Confusion2Vec Representation Framework. The proposed subword modeling is presented in Section Confusion2Vec 2.0 subword model. Section Evaluations gives details of the evaluation techniques employed for assessing the word embedding models. The experimental setup and results of various analogy and similarity tasks are presented in section Analogy & Similarity Tasks. Section Spoken Language Intent Detection presents the application of the proposed word vector representation to the spoken language intent detection task. Finally, the paper is concluded in section Conclusion.

## Confusion2Vec representation framework

In psycho-acoustics, it is established that humans also relate words with how they sound [19] in addition to semantics and syntax. Inspired by principles of human speech production and perception, we previously proposed Confusion2vec [10]. The core idea is to estimate a hyper-space that not only captures the semantics and syntax of human language, but also augments the vector space with acoustic ambiguity information, i.e., word acoustic similarity information. In other words, word2vec, GloVe can be viewed as a subspace of the Confusion2vec vector space.

Several different methodologies are proposed in [10] for capturing the ambiguity information. The methodologies are an adaptation of the skip-gram modeling for word confusion networks or lattice-like structures. The word lattices are directed acyclic weighted graphs of all the word sequences that are likely possible. A confusion network is a specific type of lattice with constraints that each word sequence passes through each node of graph. Such lattice-like structures can be derived from machine learning algorithms that output probability measures, for example, an ASR. Fig 1, illustrates a confusion network that can possibly result from a speech recognition system. Unlike typical simple sentences which are used for training word embeddings like word2vec, GloVe, BERT, ELMo etc., the information in the confusion network can

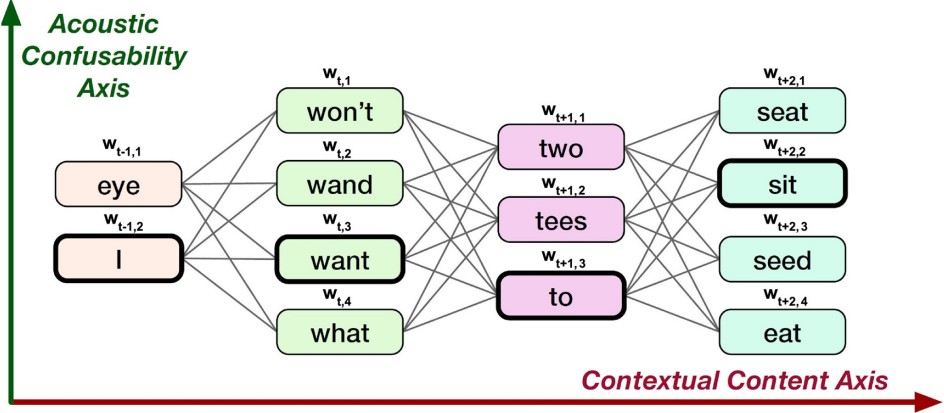

**Fig 1. Example confusion network output by ASR for the ground-truth phrase—"I want to sit".** Figure adapted from P. G. Shivakumar and P. Georgiou, *"Confusion2vec: Towards enriching vector space word representations with representational ambiguities,"* PeerJ Computer Science, vol. 5, p. e195, 2019 [10].

be viewed along two dimensions: (i) contextual dimension, and (ii) acoustic ambiguity dimension.

More specifically, four different configurations of skip-gram modeling algorithms are proposed in our previous work [10], namely: (i) top-confusion, (ii) intra-confusion, (iii) inter-confusion, and (iv) hybrid model. The top-confusion version considers only the most-probable path of the ASR confusion network and applies the typical skip-gram model on it. The intra-confusion version applies the skip-gram modeling on the acoustic ambiguity dimension of the confusion network and ignores the contextual information, i.e., each ambiguous word alternative is predicted by the other over a pre-defined local context. The inter-confusion version applies the skip-gram modeling on the contextual dimension but over each of the acoustic ambiguous words. The hybrid model is a combination of both the intra and inter-confusion configurations. More information on the training configuration is available in [10]. The present work builds upon this basic Confusion2vec framework.

## Confusion2Vec 2.0 subword model

Subword encoding of words has been popular in modeling semantics and syntax of language using word vector representations [6, 8, 9]. The use of subwords are mainly motivated by the fact that the subwords incorporate morphological information which can be helpful, for example, in relating the prefixes, suffixes and the word root. In this work, we apply subword representation for encoding the word ambiguity information in the human language. We believe we have a compelling case for the use of subwords for representing the acoustic similarities (ambiguities) between the words in the language for the following reasons. Similar sounding words often tend to have similar set of characters and thus have a high degree of overlapping set of character n-grams. This subword based feature encoding helps model the level of overlap and estimate the magnitude of acoustic similarity robustly. Moreover, the use of subwords should help in efficient encoding of under-represented words in the language [9]. This is crucial in the case of Confusion2vec because we are restricted to speech data and their corresponding decoded ASR lattices for training which leads to data sparsity issues compared to computing typical word vector representation that can be trained on large amounts of easily available plain text data. Another important aspect is the ability to represent out-of-vocabulary words [18] which are common with end-to-end ASR systems outputting character sequences.

In the proposed model, for example, each word $w$ is represented as a sum of its constituent n-gram character subwords. This enables the model to infer the internal structure of each word. For example, a word "`want`" is represented with the vector sum of the following subwords:

$$< \texttt{wa, wan, ant, nt} >, \ < \texttt{wan, want, ant} >, \ < \texttt{want, want} >, \ < \texttt{want} >$$

Symbols $<$ and $>$ are used to represent the beginning and end of the word. The n-grams are generated for n = 3 up to n = 6. The choice of length of character n-grams is language dependent and empirically chosen for English [9]. It is apparent that an acoustically ambiguous, similar sounding word "`wand`" has a high degree of overlap with the set of n-gram characters.

In this paper, we consider two modeling variations: (i) inter-confusion, and (ii) intra-confusion versions of Confusion2vec with the subword encoding.

## Intra-Confusion model

The goal of the intra-confusion model is to estimate the inter-word relations between the acoustically ambiguous words that appear in the ASR lattices. For this, we perform skip-gram modeling over the acoustic similarity dimension (see Fig 1) and ignore the contextual dimension of the utterance. The objective of the intra-confusion model is to maximize the following log-likelihood:

$$\sum_{t=1}^{T} \sum_{\hat{a} \in \hat{A}_t} \sum_{a \in A_t} log \ p(w_{t,a} | w_{t,\hat{a}}) \tag{1}$$

where $T$ is the length of the utterance (confusion network) in terms of number of words, $w_{i,j}$ is the word in the confusion network output by the ASR at time-step $i$ and $j$ is the index of the word among the ambiguous alternatives. $\hat{A}_t$ is the set of indices of all ambiguous words at time-step $t$, $\hat{a}$ is the index of the current word along the acoustic ambiguity dimension, $A_t \subseteq \hat{A}_t - \hat{a}$ is the subset of ambiguous words barring $\hat{a}$ at the current word $t$, i.e., for example from Fig 1, for the current word, $w_{t,\hat{a}}$, "`want`", $A_t \subseteq \{\texttt{wand, won't, what}\}$. Additionally, for subword encoding, each word input is represented as:

$$w_{i,j} = \sum_{s \in S_w} x_s \tag{2}$$

where $S_w$ is the set of all character n-grams ranging from n = 3 to n = 6 and the word itself and $x_s$ is the vector representation for n-gram subword $s$. Few training samples (input, target) generated for this configuration pertaining to input confusion network in Fig 1 are (`I, eye`), (`eye, I`), (`want, wand`), (`want, won't`), (`won't, what`), (`wand, what`) etc.

## Inter-Confusion model

The aim of the inter-confusion model is to jointly model the contextual co-occurrence information and the acoustic ambiguity co-occurrence information along both the axes depicted in the confusion network. Here, the skip-gram modeling is performed over time context and over all the possible acoustic ambiguities. The objective of the inter-confusion model is to maximize the following log-likelihood:

$$\sum_{t=1}^{T} \sum_{\hat{a} \in \hat{A}_t} \sum_{c \in C_t} \sum_{a \in A_c} log \ p(w_{c,a} | w_{t,\hat{a}}) \tag{3}$$

where $C_t$ corresponds to set of indices of nodes of confusion network, i.e., words around the current word $t$ along the time-axis and $c$ is the current context index. $A_c$ is the set of indices of acoustically ambiguous words at a context $c$. For example, for the current word, $w_{t,\hat{a}}$, "want" in Fig 1, $A_c \subseteq$ {I, eye, two, tees, to, seat, sit, seed, eat} and $A_t \subseteq$ {wand, won't, what, want}. Note, each word input is subword encoded as in Eq 2. Few training samples (input, target) generated for this configuration are (want, I), (want, eye), (want, two), (want, to), (want, tees), (what, I), (what, eye), (what, to), (what, tees), (what, two), (won't, eye) etc.

## Training loss and objective

Negative sampling is employed for training the embedding model. Negative sampling was first introduced for training word2vec representation [4]. It is a simplification of the Noise Contrastive Estimation objective [20]. The negative sampling for training the embedding can be posed as a set of binary classification problems which operates on two classes: presence of signal or absence (noise). In the context of word embeddings the presence of the context words are treated as positive class and the negative class is randomly sampled from the unigram distribution of the vocabulary. The negative sampling loss function to be optimized for subword model can be expressed as:

$$J(\theta) = log\ \sigma\left(\sum_{s \in S_{w_i}} x_s^T o_{w_t}\right) + \sum_{k=1}^{K} \mathbb{E}_{w_k \sim P_n(w)} log\ \sigma\left(-\sum_{s \in S_{w_i}} x_s^T o_{w_k}\right) \tag{4}$$

where $\sigma(x) = \frac{1}{1+e^{-x}}$, $w_i$ is the input word, $w_t$ is the output word, $S_{w_i}$ is the set of n-gram character subwords for the word $w_i$, $x_s$ is the vector representation for the character n-gram subword $s$ and $o_{w_t}$ is the output vector representation of target word $w_t$. $K$ is the number of negative samples to be drawn from the noise distribution $P_n(w)$. The noise distribution $P_n(w)$ is chosen to be the unigram distribution of words in the vocabulary raised to the 3/4th power as suggested in [4]. Note, for Confusion2vec the input word $w_i$ and target word $w_t$ are derived according to Eqs 1 and 3 for implementing the respective training configurations.

## Evaluations

We perform evaluations of the proposed word embeddings along two aspects. One, assessing the useful, meaningful information embedded in the word vector representation. Second, in its application to a realistic task of spoken language intent detection. Note, all the evaluations, analysis and databases used in this work are in the English language.

## Analogy and similarity tasks

For evaluating the inherent semantic and syntactic knowledge of the word embeddings, we employ two tasks: (i) the semantic-syntactic analogy task, and (ii) the word similarity task.

**Semantic-Syntactic analogy task.** The word analogy task was first proposed in [3] which comprises word pair analogy questions of the form $W_1$ is to $W_2$ as $W_3$ is to $W_4$. For example, "Boy" is to "Girl" as "Son" is to "Daughter". The analogy is answered correctly if $vec(W_1) - vec(W_2) + vec(W_3)$ is most similar to $vec(W_4)$. The task comprises 19,544 analogy questions as originally compiled and released by [3]. We prune the analogy question database to match the training dataset vocabulary and to obtain identical setup used in [10] for comparison purposes.

**Word similarity task.** Another prominent approach in the NLP community [5, 21] for evaluating word vector representations is the word similarity task. We use the WordSim-353 database [22] consisting of 353 pairs of words manually annotated over a score of 1 to 10 depending on the magnitude of word similarity as perceived by humans. The task involves computing the rank-correlation (Spearman correlation) between the human annotated scores and the cosine similarity of the corresponding word vector pairs [21]. A high correlation indicates that the word vector representation captures the word similarity order similar to that perceived by humans.

For assessing the word acoustic ambiguity (similarity) information, we conduct the Acoustic analogy task, Semantic&syntactic–acoustic analogy task and Acoustic similarity tasks, all proposed in [10].

**Acoustic analogy task.** The Acoustic analogy task comprises word pair analogies compiled using homophones which answer questions of the form: $W_1$ sounds similar to $W_2$ as $W_3$ sounds similar to $W_4$. For example, "Boy" sounds similar to "Buoy" as "Sun" sounds similar to "Son". The task comprises 2,678 analogy questions and is designed to assess the ambiguity information embedded in the word vector space [10].

**Semantic&Syntactic-Analogy task.** The semantic&syntactic-acoustic analogy task is designed to assess semantic, syntactic and acoustic ambiguity information simultaneously. The analogies are formed by replacing certain words by their homophone alternatives in the original semantic and syntactic analogy task [10]. For example, "Boy" is to "Girl" as "Sun" is to "Daughter". The task comprises 3860 analogy questions. Examples of the analogies can be found in [10].

**Acoustic word similarity task.** The acoustic word similarity task is analogous to the word similarity task, i.e., it contains 943 word pairs which are rated on their acoustic similarity based on the normalized phone edit distances. A value of 1.0 refers to two words sounding identical and 0.0 refers to the word pairs being acoustically dissimilar. The task involves computing the rank-correlation (Spearman correlation) between the normalized phone edit distances and the cosine similarity of the corresponding word vector pairs.

More details regarding the evaluation methodologies are available in [10]. The evaluation datasets are made available at https://github.com/pgurunath/confusion2vec_2.0. Note, for evaluation of Confusion2vec models with analogy tasks, we compute accuracy over top-2 nearest vectors, i.e., we count the analogy as answered correctly if any of the top-2 nearest vectors satisfies the analogy. This is because, (i) the 3 analogy tasks are not mutually exclusive, and (ii) the nearest vector query with Confusion2vec, can be either along the contextual axis (semantics/syntax) or along the acoustic similarity axis. However, in case of baseline models including fastText and word2vec (W2V), we conduct typical evaluation with nearest vector (top-1) since they model only the contextual information. However, we provide both the results of top-1 and top-2 evaluations in S1 Appendix for the benefit of the reader. More information regarding evaluation can be found in [10].

## Spoken language intent classification

We also evaluate the efficacy of the proposed word representation models on the task of spoken language intent classification. A recurrent neural network (RNN) based classifier is employed by initializing the embedding layer with the proposed word vectors. Classification experiments are conducted by training the recurrent neural network on (i) clean manual transcripts, and (ii) noisy ASR transcripts, with evaluations on both manual and ASR transcripts. Classification error rates of the intent detection is used to derive assessments of the word vector representations.

## Analogy & similarity tasks

### Database

The Fisher English Training Part 1, Speech (LDC2004S13) and Fisher English Training Part 2, Speech (LDC2005S13) corpora [23] are used for both training the ASR and the Confusion2vec 2.0 embeddings. The choice of database is based on [10] for direct comparison purposes. The corpus consists of spontaneous telephonic conversations between 11,972 native English speakers. The speech data amounts to approximately 1,915 hours sampled at 8 kHz. The corpus is divided into 3 parts for training (1,905 hours, 1,871,731 utterances), development (5 hours, 5000 utterances) and test (5 hours, 5000 utterances). Overall, the transcripts contain approximately 20.8 million word tokens with 42,150 unique words.

### Experimental setup

The experimental setup is maintained identical to [10] for direct comparison. Brief detail of the setup is as follows:

**Automatic speech recognition.** A hybrid HMM-DNN based acoustic model is trained on the train subset of the speech corpus using the KALDI speech recognition toolkit [24]. 40 dimensional mel frequency cepstral coefficients (MFCC) features are extracted along with the i-vector features for training the acoustic model. The i-vector features are used to provide speaker and channel characteristics to aid acoustic modeling. The DNN acoustic model, comprises 7 layers with P-norm non-linearity (p = 2) each with 350 units [25]. The DNN is trained using 5 MFCC frame splices with left and right context of 2 to classify among 7979 Gaussian mixtures with stochastic gradient descent optimizer. The CMU pronunciation dictionary [26] is used as the word-pronunciation transcription lexicon. A tri-gram language model is trained on the training subset of the Fisher English Speech Corpus. The ASR yields word error rates (WER) of 16.57% and 18.12% on the development and the test datasets. Lattices are derived during the ASR decoding with a decoding beam size of 11 and lattice beam size of 6. The lattices are converted to confusion networks with the minimum Bayes risk criterion [27] for training the Confusion2vec embeddings. The resulting confusion networks have a vocabulary size of 41,274 and 69.5 million words, with an average of 3.34 alternative (ambiguous) words for each edge in the graph.

**Confusion2Vec 2.0.** In order to train the embedding, most frequent words are sub-sampled as suggested in [4], with the rejection threshold set to $10^{-4}$. Also, a minimum frequency threshold of 5 is set and the rarely occurring words are pruned from the vocabulary. The context window size for both the acoustic ambiguity and contextual dimensions are uniformly sampled between 1 and 5. The dimension of the word vectors is set to 300. The number of negative samples for negative sampling is chosen to be 64. The learning rate is set to 0.01 and trained for a total of 15 epochs using stochastic gradient descent. All the hyperparameters are empirically chosen for optimal performance on the development set. We implemented the Confusion2vec 2.0 by modifying the source code from fastText [9, 28]. We make our source code and trained models available at https://github.com/pgurunath/confusion2vec_2.0. The models were trained on CPU only with the following machine configuration: dual Intel Xeon CPU E5–2670 operating at 2.6GHz based on 64bit architecture. The machine had a total of 32 threads, i.e., each CPU comprises 8 cores with 2 threads per core. The machine was equipped with 128 GB of DDR3 memory, i.e., 8 x 8GB memory per CPU. With the above machine configuration and the above mentioned experimental setup, training intra-confusion model took approximately 46 minutes and inter-confusion model took approximately 3 hours and 24 minutes.

## Results

Table 1 lists the results in terms of accuracies for analogy tasks and rank-correlations for similarity tasks. The first two rows correspond to results with the original word2vec. Google W2V model is the open source model released by Google [29], trained on 100 billion word Google News dataset. The fastText model employed is the open source model trained on Wikipedia dumps with a vocabulary size of more than 2.5 million words released by Facebook [30]. We also train an in-domain version of original word2vec and fastText on the Fisher English corpus manual transcriptions for fair comparison with the Confusion2vec models, referred to as "In-domain W2V" in Table 1. C2V-1 refers to top-confusion scheme [10], which is roughly equivalent to training skip-gram model on the ASR transcripts of the Fisher English corpus. The middle three rows of the table correspond to Confusion2vec embeddings without subword encoding and they are taken directly from [10]. The bottom three rows correspond to the results obtained with subword encoding. Note, the Confusion2vec 1.0 is initialized on the Google word2vec model for better convergence. The Confusion2vec 2.0 model is initialized on the fastText model to maintain compatibility with subword encodings. We normalize the vocabulary for all the experiments, meaning the same vocabulary is used to evaluate the analogy and similarity tasks to allow for fair comparisons.

Comparing the baseline word2vec and fastText embeddings to the Confusion2vec, we observe the baseline embeddings perform well on the semantic&syntactic analogy task and provide good positive correlation on the word similarity task as expected. However, they perform poorly on the acoustic analogy task, semantic&syntactic-acoustic analogy task and give small negative correlation on the acoustic analogy task. The in-domain word2vec model undergoes a significant dip in correlation evaluating for word similarity task (0.6893 to 0.4417). Similarly, the in-domain version of the fastText also sees degradation of correlation from 0.7361 to 0.3584. We believe the limited data and restricted vocabulary of the in-domain versions are responsible for the degradation. We also note that the subword encoding in fastText models is particularly more susceptible. All the Confusion2vec models perform relatively well on the semantic&syntactic analogy task and word similarity task, but more importantly,

**Table 1. Results: Different proposed models.**

| | Model | Analogy Tasks | | | | Similarity Tasks | |
|---|---|---|---|---|---|---|---|
| | | S&S | Acoustic | S&S-Acoustic | Average Accuracy | Word Similarity | Acoustic Similarity |
| | Google W2V [4] | 61.42% | 0.9% | 16.99% | 26.44% | **0.6893** | -0.3489 |
| | In-domain W2V | 59.17% | 0.6% | 8.15% | 22.64% | 0.4417 | -0.4377 |
| | fastText [9] | **75.93%** | 0.46% | 17.40% | 31.26% | 0.7361 | -0.3659 |
| | In-domain fastText | 46.45% | 0.75% | 17.05% | 21.42% | 0.3584 | 0.2610 |
| **Confusion2Vec 1.0** (word) [10] | C2V-1 | 70.56% | 1.46% | 23.86% | 31.96% | 0.6036 | -0.4327 |
| | C2V-a | 63.97% | 16.92% | 43.34% | 41.41% | 0.5228 | 0.6200 |
| | C2V-c | **65.45%** | 27.33% | 38.29% | 43.69% | 0.5798 | 0.5825 |
| **Confusion2Vec 2.0** (subword) | C2V-1 | 56.83% | 1.46% | 20.99% | 26.43% | 0.3720 | 0.3022 |
| | C2V-a | 56.74% | **50.79%** | **44.67%** | **50.73%** | 0.2929 | **0.8108** |
| | C2V-c | 56.87% | **51.00%** | **44.98%** | **50.95%** | 0.2893 | **0.8106** |

C2V-a: Intra-Confusion; C2V-c: Inter-Confusion; C2V-1: Top-Confusion; S&S: Semantic & Syntactic Analogy.

The results of the analogy tasks represent percentage accuracy; and the results of the similarity tasks represent Spearman correlation.

For the analogy tasks: the accuracies of baseline word2vec, fastText models are for top-1 evaluations, whereas of the other models are for top-2 evaluations (as discussed in [10]). For the similarity tasks: all the correlations (Spearman's) are statistically significant with $p < 0.001$. See Tables 6 and 8 in S1 Appendix for more detailed results.

yield high accuracies on acoustic analogy task and semantic&syntactic-acoustic analogy tasks and provide high positive correlation with the acoustic similarity task.

Comparisons between Confusion2vec 1.0 and Confusion2vec 2.0 among the analogy tasks reveal the subword encoding enhances the acoustic ambiguity modeling. For the acoustic analogy task we find relative improvement of up to 46.41% over its non-subword counterpart. Moreover, even for the semantic&syntactic-acoustic analogy task, we observe improvements with subword encoding. However, we find a small reduction in performance for the original semantic and syntactic analogy task. One explanation for this is that the different analogy tasks are fairly, mutually exclusive, i.e., getting right on one task compromises performance on the other. The top-2 evaluations for Confusion2Vec provides a partial solution to this. Nevertheless, there can be instances where the embedding can favor information on either acoustic ambiguity or contextual information dimension. Thus, there exists trade-off between the different proposed analogy based evaluation tasks. The goal is to optimize this trade-off as best as possible. One way to judge this trade-off is to look at the average accuracy across the analogy tasks. Regardless of the small dip in the performance, the accuracies remain acceptable in comparison to the in-domain word2vec and fastText models. Overall, taking the average accuracy of all the analogy tasks, we obtain an increase of approximately 16.62% relative over the non-subword Confusion2vec models.

Investigating the results for the similarity tasks, we find a significant correlation of 0.81 for acoustic similarity task with the subword encoding. However, again, a small degradation is observed with the word similarity task obtaining a correlation of 0.2929 against the 0.3584 of the in-domain baseline fastText model. In contrast to Confusion2vec 2.0, Confusion2vec 1.0 is able to improve correlation on word similarity task against its counterpart in-domain word2-vec model (from 0.4417 to 0.5798). Our investigations on the possible causes of this lower correlation on the word similarity task reveals the following: First, the same set of word pairs are scored for both the word similarity and acoustic similarity tasks, and thus increase in the performance of one similarity task resulting in slight compromise on the other is inevitable. Second, we found that in case of Confusion2vec 2.0, the Pearson correlation was always higher than the Spearman correlation (see Table 8 in S1 Appendix). This likely suggests that with Confusion2vec 2.0 models, while the linearity especially at the tails of the distribution is relatively stronger, the monotonicity is negatively impacted particularly at and around the mean. We believe this is a fair compromise since we are more concerned of words that are more similar or more dissimilar to the others and less concerned of neutral words. Finally, this is also supported by the results on original analogy task which performs fairly well (concerned with the most similar word). Overall, we believe that the subword modeling with Confusion2vec enhances the acoustic confusability modeling considerably, and this causes slight disruptions in semantic modeling while preserving the important and relevant semantics of the language.

## Model concatenation

Further, the Confusion2vec model can be concatenated with the other word embedding models to produce a new word vector space that can result in better representations as seen in [10]. Table 2 lists the results of the concatenated models. For the previous, non-subword version of the Confusion2vec, the vector models are concatenated with the word2vec model trained on the ASR output transcripts (C2V-1). The choice of using the C2V-1 instead of the Google W2V for concatenation was based on empirical findings. Where as to maintain compatibility of subword encoding, the Confusion2vec 2.0 models are concatenated with fastText models. Note the Confusion2vec 1.0 C2V-1 is pre-trained on Google's W2V model for fair comparison against concatenation of Confusion2vec 2.0 models with fastText.

**Table 2. Results: Concatenated models.**

| | Model | Analogy Tasks | | | | Similarity Tasks | |
|---|---|---|---|---|---|---|---|
| | | S&S | Acoustic | S&S-Acoustic | Average Accuracy | Word Similarity | Acoustic Similarity |
| | Google W2V [4] | 61.42% | 0.9% | 16.99% | 26.44% | **0.6893** | -0.3489 |
| | In-domain W2V | 59.17% | 0.6% | 8.15% | 22.64% | 0.4417 | -0.4377 |
| | fastText [9] | 75.93% | 0.46% | 17.40% | 31.26% | 0.7361 | -0.3659 |
| **Confusion2Vec 1.0** (word) [10] | C2V-1 + C2V-a | 67.03% | 25.43% | 40.36% | 44.27% | 0.5102 | 0.7231 |
| | C2V-1 + C2V-c | 70.84% | 35.25% | 35.18% | 47.09% | 0.5609 | 0.6345 |
| | C2V-1 + C2V-c (UJO) | 65.88% | **49.4%** | 41.51% | **52.26%** | 0.5379 | **0.7717** |
| **Confusion2Vec 2.0** (subword) | fastText + C2V-a | **76.10%** | 22.67% | **49.15%** | 49.31% | 0.5744 | **0.7577** |
| | fastText + C2V-c | **76.16%** | 22.56% | **49.12%** | 49.12% | 0.5732 | **0.7573** |

C2V-a: Intra-Confusion; C2V-c: Inter-Confusion; C2V-1: Top-Confusion; S&S: Semantic & Syntactic Analogy; UJO: Unrestricted Joint Optimization (see [10]). The results of the analogy tasks represent percentage accuracy; and the results of the similarity tasks represent Spearman correlation. For the analogy tasks: the accuracies of baseline word2vec, fastText models are for top-1 evaluations, whereas of the other models are for top-2 evaluations (as discussed in [10]). For the similarity tasks: all the correlations (Spearman's) are statistically significant with $p < 0.001$. See Tables 7 and 9 in S1 Appendix for more detailed results.

First, comparisons between the non-concatenated versions in Table 1 and the concatenated version in Table 2, of the non-subword models, we observe an improvement of approximately 7.22% relative in average analogy accuracy after concatenation. We don't observe significant improvement with subword based models after concatenation in terms of average analogy accuracy. However, we observe different dynamics between the acoustic ambiguity and the semantic and syntactic subspaces. Concatenation results in improved semantic and syntactic evaluations at the expense of degradation in accuracies of acoustic analogy task. We also note improvements (9.27% relative) in semantic&syntactic-acoustic analogy task after concatenation, confirming meaningful existence of both ambiguity and semantic-syntactic relations. Moreover, concatenation also yields a better correlation on the word similarity task.

Next, comparisons of the Confusion2vec 1.0 (non-subword) and the subword version, we observe significant improvements in the semantic&syntactic analogy task (7.51% relative) as well as the semantic&syntactic-acoustic analogy tasks (21.78% relative). Moreover, the subword models outperform the non-subword version in both of the similarity tasks. The subword models slightly under-perform in the acoustic analogy task, but more crucially outperform the Google W2V and FastText baselines significantly. Overall, these changes in dynamics between the acoustic and semantic/syntactic subspaces observed in the case of concatenated models can be attributed to the fact that we are optimizing a different criterion than the non-concatenated versions.

Further, the concatenated models can be fine-tuned and optimized to exploit additional gains as found in [10]. The row corresponding to Confusion2Vec 1.0 − C2V-1 + C2V-c (UJO) is the best result obtained in [10] which involves 2-passes. The Confusion2Vec 2.0 with the subword modeling with a single pass training gives comparable performance to the 2-pass approach. Thus we skip the 2-pass approach with the subword model in favor of ease of training and reproducibility.

## Embedding visualization

Fig 2 illustrates the word vector spaces of fastText embeddings and the proposed C2V-a embeddings after dimension reduction using principal component analysis. The visualizations

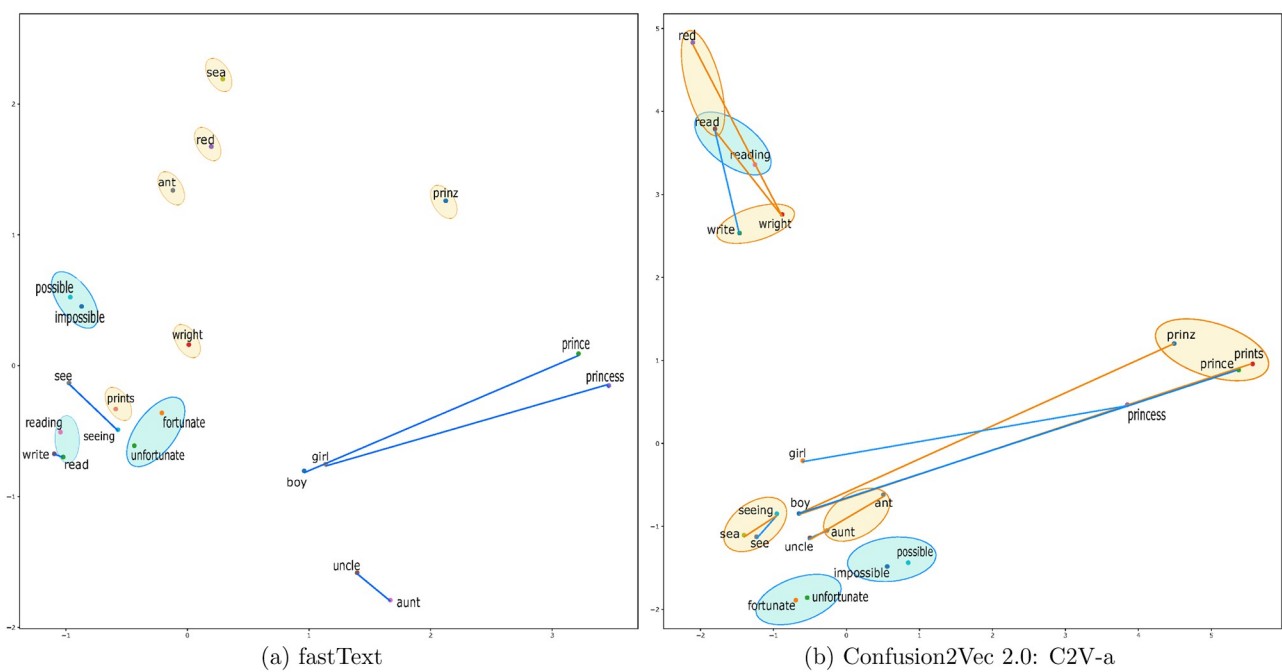

**Fig 2. 2-D plots of selected word vectors portraying semantic, syntactic and acoustic relationships after dimension reduction using PCA.** The blue lines indicate semantic relationships, blue ellipses indicate syntactic relationships, orange lines indicate acoustic-semantic/syntactic relations and orange ellipses indicate acoustic ambiguity word relations. Plots with identical word sets corresponding to Confusion2Vec 1.0 and Google W2V can be found in [11]. Please note that the out-of-vocabulary word "prinz" cannot be represented in Google W2V and Confusion2Vec 1.0 spaces.

are generated using scikit-learn and matplotlib python packages. We observe meaningful interactions between the semantic&syntactic subspace and the acoustic ambiguity subspace. For example, in Fig 2, vectors *"boy"-"prince"*, *"see"-"seeing"*, *"read"-"write"*, *"uncle"-"aunt"* are similar to acoustically ambiguous vector *"boy"-"prints"*, *"sea"-"seeing"*, *"read"-"write"*, *"uncle"-"ant"* respectively which is not the case in Fig 2 with fastText embeddings. Such vector relationships can be exploited for downstream spoken language applications by providing crucial acoustic ambiguity information to recover from speech recognition errors. Also note, the acoustically ambiguous words such as *"prinz"*, *"prince"*, *"prints"* are found clustered together. Another important observation is that the word *"prinz"*, out-of-vocabulary in English, has an orphaned representation under fastText in Fig 2. However, *"prinz"* finds a meaningful representation on the basis of acoustic signature in the proposed Confusion2vec model as seen in Fig 2, i.e., *"prinz"* is clustered together with acoustically similar words *"prince"* & *"prints"* and the vector *"boy"-"prinz"* is similar to vector *"boy"-"prince"*. Occurrence of out-of-vocabulary words such as *"prinz"* is commonplace with end-to-end ASR systems that output characters prone to errors. Note, out-of-vocabulary words such as *"prinz"* cannot be represented by typical word embeddings such as word2vec, GloVe, etc., and hence sub-optimal for representation with many end-to-end ASR systems.

## Spoken language intent detection

In this section, we apply the proposed word vector embedding to the task of spoken language intent detection. Spoken language intent detection is the process of decoding the speaker's intent in contexts involving voice commands, call routing and any human computer interactions. Many spoken language technologies use an ASR to convert the speech signal to text, a process prone to errors due to the varying speakers and noisy environments. The erroneous

ASR outputs in turn result in degradation of the downstream intent classification. Few efforts have focused on handling the errors of the ASR to make the subsequent intent detection process more robust to errors. These efforts often involve training the intent classification systems on noisy ASR transcripts. The downsides of training the intent classifiers on the ASR transcripts is that the systems are limited to the amount of speech data available. Moreover, varying speech signal conditions and use of different ASR models make such classifiers non-optimal and less practical. In many scenarios, speech data is not available to enable adaptation on ASR transcripts.

In our previous work [11], we applied the non-subword version of the Confusion2vec to the task of spoken language intent detection. We demonstrated that the Confusion2vec is able to perform as efficiently as the popular word embeddings like word2vec and GloVe on clean manual transcripts, giving comparable classification error rates. More importantly, we were able to illustrate the robustness of the Confusion2vec embeddings when evaluated on the noisy ASR transcripts, resulting in up-to relative 20% improvements. We showed that the results also translate to models trained on noisy ASR transcripts.

In this paper, we incorporate the Confusion2vec 2.0 embeddings and exploit the enhanced effects of the subword modeling in capturing acoustic ambiguity as verified by the previous evaluations in Section Analogy & Similarity Tasks. We believe the proposed model could further improve and provide robustness to the spoken utterance classification and thereby, aim to eliminate the need for re-training the classifiers on the ASR outputs.

## Database

We conduct experiments on the Airline Travel Information Systems (ATIS) benchmark dataset [31]. The dataset consists of humans making flight-related inquiries in the English language with an automated answering machine with audio recorded and its transcripts manually annotated. ATIS consists of 18 intent categories. The dataset is divided into train (4478 samples), development (500 samples) and test (893 samples) consistent with previous works [11, 32, 33]. For ASR evaluations, the audio recordings are down-sampled from 16kHz to 8kHz and then decoded using the ASR setup described in section Automatic speech recognition using the audio mappings provided in https://github.com/pgurunath/slu_confusion2vec. The ASR achieves a WER of 18.54% on the ATIS test set.

## Experimental setup

For intent classification we adopt a simple RNN architecture identical to [11]. This allows for direct comparison. The architecture of the neural network is intentionally kept simple in order to assess the efficacy of the proposed embedding word features. The classifier is comprised of an embedding layer followed by a single layer of bi-directional recurrent neural network (RNN) with long short-term memory (LSTM) units. This is followed by a linear dense layer and softmax function. The softmax outputs a probability distribution across all the intent categories. The embedding layer is fixed throughout the training. However, in the case of the randomly initialized embeddings, the embedding is estimated on the in-domain data used for intent detection.

The intent classification models are trained on the 4478 samples of training subset and the hyperparameters are tuned on the development set. We choose the set of hyperparameters yielding the best results on the development set and then apply it on the unseen held-out test subset. The results are reported on both the manual clean transcripts and the ASR transcripts. For training we treat each utterance as a single sample (batch size = 1). The hyper-parameter space we experiment are as follows: the hidden dimension size of the LSTM is tuned over {32,

64, 128, 256}, the learning rate over {0.0005, 0.001}, and the dropout is tuned over {0.1, 0.15, 0.2, 0.25}. The Adam optimizer is employed for optimization and trained for a total of 50 epochs. We employ early stopping when the loss on the development set doesn't improve for 5 consecutive epochs.

## Baselines

We include results from several baseline systems for providing comparisons of Confusion2Vec 2.0 with the popular context-free word embeddings, contextual embeddings, established NLU systems and the current state-of-the-art.

1. **Context-Free Embeddings**: GloVe [5, 34], skip-gram word2vec [4, 29] and fastText [9, 30] word representations are employed. They are referred to as context-free embeddings since the word representations are static irrespective of the context.

2. **ELMo**: [6] proposed deep contextualized word representation based on character based deep bidirectional language model trained on large text corpus. ELMo effectively models syntax and semantics of the language along varying linguistic contexts. Unlike context-free embeddings, ELMo embeddings have varying representations for each word depending on the word's context. We employ the original model trained on 1 Billion Word Benchmark with 93.6 million parameters [35]. For intent-classification we add a single bi-directional LSTM layer with attention. We experiment with two versions of the model, one with intent classification only and the other with multi-task joint intent and slot predictions.

3. **BERT**: [8] introduced BERT—bidirectional contextual word representations based on self attention mechanism of Transformer models. BERT models make use of masked language modeling and next sentence prediction to model language. Similar to ELMo, the word embeddings are contextual, i.e., varying according to the context. We employ "bert-base-uncased" model [36] with 12 layers of 768 dimensions each trained on BookCorpus and English Wikipedia corpus. For intent-classification we add a single bi-directional LSTM layer with attention. We experiment with two versions of the model, one with intent classification only and the other with multi-task joint intent and slot predictions.

4. **Joint SLU-LM**: [37] employed joint modeling of the next word prediction along with intent and slot labeling. The unidirectional RNN model updates intent states for each word input and uses it as context for slot labeling and language modeling.

5. **Attn. RNN Joint SLU**: [38] proposed attention based encoder-decoder bidirectional RNN model in a multi-task model for joint intent and slot-filling tasks. A weighted average of the bidirectional LSTM hidden states of the encoder network provides information from parts of the input word sequence which is used together with time aligned encoder hidden state for the decoder to predict the slot labels and intent.

6. **Slot-Gated Attn**: [33] introduced a slot-gated mechanism which introduces additional gate to improve slot and intent prediction performance by leveraging intent context vector for slot filling task.

7. **Self Attn. SLU**: [39] proposed self-attention model with gate mechanism for joint learning of intent classification and slot filling by utilizing the semantic correlation between slots and intents. The model estimates embeddings augmented with intent information using self attention mechanism which is utilized as a gate for slot filling task.

8. **Joint BERT**: [40] proposed to use BERT embeddings for joint modeling of intent and slot-filling. The pre-trained BERT embeddings are fine-tuned for (i) sentence prediction task—

intent detection, and (ii) sequence prediction task—slot filling. The Joint BERT model lacks the bi-directional LSTM layer in comparison to the earlier baseline *BERT* based model.

9. **SF-ID Network**: [41] introduced a bi-directional interrelated model for joint modeling of intent detection and slot-filling. An iteration mechanism is proposed where the SF subnet introduces the intent information to slot-filling task while the ID-subnet applies the slot information to intent detection task. For the task of slot-filling, a conditional random field layer is used to derive the final output.

10. **ASR Robust ELMo**: [17] proposed ASR robust contextualized embeddings for intent detection. ELMo embeddings are fine-tuned with a novel loss function which minimizes the cosine distance between the acoustically confused words found in ASR confusion networks. Two techniques based on supervised and unsupervised extraction of word confusions are explored. The fine-tuned contextualized embeddings are then utilized for spoken language intent detection.

## Results

In this section, we conduct experiments by training models on (i) clean human annotations and (ii) noisy ASR transcriptions.

**Training on clean transcripts.** Table 3 lists the results of the intent detection in terms of classification error rates (CER). The "Reference" column corresponds to results on human transcribed ATIS audio and the "ASR" corresponds to the evaluations on the noisy speech recognition transcripts. Firstly, evaluating on the Reference clean transcripts, we observe the Confusion2vec 2.0 with subword encoding is able to achieve the third best performance. The best-performing Confusion2vec 2.0 achieves a CER of 1.79%. Among the different versions of the proposed subword based Confusion2vec, we find that the concatenated versions are better. We believe this is because the concatenated models exhibit better semantic and syntactic relations (see Tables 1 and 2) compared to the non-concatenated ones. Among the baseline models, the contextual embedding like BERT and ELMo gives the best CER. Note, the proposed Confusion2vec embeddings are context-free and are able to outperform other context-free embedding models such as GloVe, word2vec and fastText.

Secondly, evaluating the performance on the noisy ASR transcripts, we find that all the subword based Confusion2vec 2.0 models outperform the popular word vector embeddings by a big margin. The subword-Confusion2vec gives an improvement of approximately 45.78% relative to the best performing context-free word embeddings. The proposed embeddings also improve over the contextual embeddings including BERT and ELMo (relative improvements of 29.06%). Moreover, the results are also an improvement over the non-subword Confusion2-vec word vectors (31.50% improvement). Comparisons between the different versions of the proposed Confusion2vec show that the intra-confusion configuration yields the least CER. The best results with the proposed model outperforms the state-of-the-art (ASR Robust ELMo [17]) by reducing the CER by a relative of 13.12%. Inspecting the degradation, $\Delta_{\text{diff}}$ (drop in performance between the clean and ASR evaluations), we find that all the Confusion2vec 2.0 with subword information undergo low degradation while giving the best CER, thereby re-affirming the robustness to noise in transcripts. This confirms that the subword encoding is better able to represent the acoustic ambiguities associated in human spoken language.

Further, analyzing the results, Table 4 lists a few examples within the domain of intent detection comparing the baseline fastText embedding and the proposed concatenated version of inter-confusion model. In the first example, the ASR incorrectly recognizes "seating" as

**Table 3. Intent Classification Error Rates (CER): Trained on clean reference transcripts, evaluated on clean reference and noisy ASR transcripts.**

| | Model | Reference | ASR | $\Delta_{\text{diff}}$ |
|---|---|---|---|---|
| Context-Free Embeddings | Random | 2.69 | 10.75 | 8.06 |
| | GloVe [5] | 1.90 | 8.17 | 6.27 |
| | Word2Vec [4] | 2.69 | 8.06 | 5.37 |
| | fastText [9] | 1.90 | 8.40 | 6.50 |
| | Joint SLU-LM [37]† | 1.90 | 9.41 | 7.51 |
| | Attn. RNN Joint SLU [38]† | 1.79 | 8.06 | 6.27 |
| | Slot-Gated Attn. [33]† | 3.92 | 10.64 | 6.72 |
| | Self Attn. SLU [39]† | 2.02 | 9.18 | 7.16 |
| | SF-ID Network [41]† | 3.14 | 10.53 | 7.39 |
| | C2V 1.0 [10] | 2.46 | 6.38 | 3.92 |
| Contextual Embeddings | ELMo [6] | 1.79 | 6.83 | 5.04 |
| | ELMo [6]† | 1.46 | 7.05 | 5.59 |
| | BERT [8] | 1.79 | 7.05 | 5.26 |
| | BERT [8]† | **1.12** | 6.16 | 5.04 |
| | Joint BERT [40]† | 2.46 | 7.73 | 5.27 |
| | ASR Robust ELMo (unsup.) [17] | 3.24 | 5.26 | 2.02 |
| | ASR Robust ELMo (sup.) [17] | 3.46 | 5.03 | **1.57** |
| Proposed Context-Free Embeddings | C2V-c 2.0 | 3.36 | 5.82 | 2.46 |
| | C2V-a 2.0 | 2.46 | **4.37** | **1.91** |
| | fastText + C2V-c 2.0 | 1.79 | **4.70** | 2.91 |
| | fastText + C2V-a 2.0 | 1.90 | **5.04** | 3.14 |

$\Delta_{\text{diff}}$ is the absolute degradation of model from clean to ASR. C2V 1.0 corresponds to C2V-1 + C2V-c (UJO) in Tables 1 and 2.

† indicates joint modeling of intent and slot-filling.

"feeding" which leads to an error in intent classification, i.e., intent is detected as "Meal" instead of "Flight Capacity". However, Confusion2Vec is able to recognize the ambiguity through better vector representation of acoustic confusions between the two unvoiced fricatives /f/ and /s/ and the consonants /d/ and /t/, phonomena that are well documented [42, 43] and eventually lead to better classification. The second example is a classic instance of homophones (fare and fair) with similar implications. In the third example, both the embeddings

**Table 4. Examples of intent detection: Trained on clean reference text, evaluated on ASR transcripts.**

| System | Text | True Intent | Predicted Intent | |
|---|---|---|---|---|
| | | | fastText | concat C2V-c 2.0 |
| Manual | "what is the seating capacity of a DC9" | Capacity | Meal | Capacity |
| ASR | "what is **to feeding** capacity of the DC9" | | | |
| Manual | "what is the lowest fare for a flight from washington dc to boston" | Airfare | Flight | Airfare |
| ASR | "what is the lowest **fair** for a flight from washington dc to boston" | | | |
| Manual | "list fares from washington dc to boston" | Airfare | Flight | Flight |
| ASR | "**lift affairs** from washington dc to boston" | | | |
| Manual | "what does fare code bh mean" | Abbreviation | Ground Service | Abbreviation |
| ASR | "**which is could be <unk> me**" | | | |

Manual refers to clean, human annotated transcripts. ASR refers to the automatic speech transcription by ASR. The bold text highlights the errors made by ASR. "concat C2V-c 2.0" refers to the proposed model: concatenated fastText + inter-confusion model

**Table 5. Intent Classification Error Rates (CER): Trained and evaluated on noisy ASR transcripts.**

| Model | WER % | CER % |
|---|---|---|
| Random | 18.54 | 5.15 |
| GloVe [5] | 18.54 | 6.94 |
| Word2Vec [4] | 18.54 | 5.49 |
| Schumann et, al, 2018 [44] | 10.55 | 5.04 |
| C2V 1.0 | 18.54 | 4.70 |
| C2V-c 2.0 | 18.54 | 4.82 |
| C2V-a 2.0 | 18.54 | **4.26** |
| fastText + C2V-c 2.0 | 18.54 | **3.70** |
| fastText + C2V-a 2.0 | 18.54 | **4.26** |

C2V 1.0 corresponds to C2V-1 + C2V-c (UJO) in Tables 1 and 2. Note, we don't domain-constrain, optimize or re-score our ASR, as in [44]

fail to recover from the error. Finally, the fourth example is a manifestation of a more complex error spanning words/phrases. The proposed Confusion2Vec is able to reconcile the acoustic ambiguity information across multiple words and successfully recognize the correct underlying intent.

**Training on noisy ASR transcripts.** Table 5 presents the results obtained by training models on the ASR transcripts and evaluated on the ASR transcripts. Here we omit all the joint intent-slot filling baseline models, since training on ASR transcript needs aligned set of slot labels due to insertion, substitution and deletion errors which is out-of-scope of this study. We note that the Confusion2vec models give significantly lower CER. The subword based Confusion2vec models also provide improvements over the non-subword based Confusion2-vec model (21.28% improvement). Comparing the results in Tables 3 and 5, we would like to highlight that the subword-Confusion2vec model gives a minimum CER of 4.37% on model trained on clean transcripts which is much better than the CER obtained by popular word embeddings like word2vec, GloVe, fastText even when trained on the ASR transcripts (15.15% better relatively). These results demonstrate that the subword-Confusion2vec models can eliminate the need for re-training the intent classification model on ASR transcripts for robust performance.

## Conclusion

In this paper, we proposed the use of subword encoding for modeling the acoustic ambiguity information and augment word vector representations along with the semantic and syntax of the language. Each word in the language is represented as a sum of its constituent character n-gram subwords. The advantages of the subword encoding are confirmed by evaluating the proposed models on various word analogy tasks and word similarity tasks designed to assess the effective acoustic ambiguity, semantic and syntactic knowledge inherent in the models. Finally, the proposed subword models are applied to the task of spoken language intent detection. The results of intent classification system suggest that the proposed subword Confusion2vec models greatly enhance the classification performance when evaluated on the noisy ASR transcripts. The results highlight that subword-Confusion2vec models are robust and domain-independent and do not need re-training of the classifier on ASR transcript.

Further, the following advantages highlight the prospects of the proposed Confusion2Vec embedding in enabling its applications in a wide range of conditions: (i) the proposed

Confusion2Vec embeddings provide feasible representations both acoustically and semantically to unseen and out-of-vocabulary words, (ii) the embedding is able to model domain independent representations in an unsupervised manner that can capture acoustic signatures of words in conjunction with semantic information, which enhances the flexibility and feasibility to train on easily available domain independent speech data, and (iii) the domain independent nature of Confusion2Vec enables cross-lingual modeling, transfer learning techniques [45, 46] for capturing ambiguous information in low-resource languages.

The proposed Confusion2Vec word embedding can benefit a range of applications involving speech (spoken language) in which acoustic ambiguity is inherent, for example in scenarios involving ASR, error correction systems, spoken language understanding, speech translation, text-to-speech systems etc. Moreover, the ambiguity need not be limited to acoustics only. Inherent ambiguities are present in various other settings dependent on the nature of the underlying signals such as for example, pictorial ambiguities associated with applications such as Optical character recognition or Image/Video Scene summarization. There is also the possibility of multiple ambiguity dimensions associated with certain applications such as Speech Translation where in addition to acoustic ambiguity, there can be ambiguity associated with source and target language morphology, segmentation and linguistic expressions such as paraphrasing. More applications are discussed in detail in [10].

In the future, we plan to model ambiguity information using deep contextual modeling techniques such as BERT. We believe bidirectional information modeling with attention can further enhance ambiguity modeling. On the application side, we plan to implement and assess the effect of using Confusion2vec models for a wide range of natural language understanding and processing applications such as speech translation, dialogue state tracking etc. We also plan to understand the factors that affect the quality of the proposed embeddings by conducting further analysis of the effects of ASR performance (WER), decoding beam size, characteristics of underlying speech signal environments including type of noise, amount of noise, channel effects, transferability over different ASR systems etc. The performance implications of these factors to the end-task are also of interest.

## Supporting information

**S1 Appendix.**
(PDF)

## Author Contributions

**Conceptualization:** Prashanth Gurunath Shivakumar.

**Formal analysis:** Prashanth Gurunath Shivakumar.

**Funding acquisition:** Shrikanth Narayanan.

**Investigation:** Prashanth Gurunath Shivakumar.

**Methodology:** Prashanth Gurunath Shivakumar.

**Project administration:** Panayiotis Georgiou, Shrikanth Narayanan.

**Resources:** Shrikanth Narayanan.

**Software:** Prashanth Gurunath Shivakumar.

**Supervision:** Panayiotis Georgiou, Shrikanth Narayanan.

**Validation:** Prashanth Gurunath Shivakumar.

**Visualization:** Prashanth Gurunath Shivakumar.

**Writing – original draft:** Prashanth Gurunath Shivakumar.

**Writing – review & editing:** Panayiotis Georgiou, Shrikanth Narayanan.

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
