## [Decision Letter · Decision Letter 0]

19 Dec 2021

PONE-D-21-33107Confusion2Vec 2.0: Enriching ambiguous spoken language representations with subwordsPLOS ONE

Dear Dr. Gurunath Shivakumar,

Thank you for submitting your manuscript to PLOS ONE. After careful consideration, we feel that it has merit but does not fully meet PLOS ONE’s publication criteria as it currently stands. Therefore, we invite you to submit a revised version of the manuscript that addresses the points raised during the review process. Both reviewers found the paper very intersting and the results significant, but there are some minor imporvments to the paper needed before it is ready for publication. I encourage the authors to carefuly read and address them.

We look forward to receiving your revised manuscript.

Kind regards,

Seyed Reza Shahamiri

Academic Editor

PLOS ONE

Journal Requirements:

3.Please review your reference list to ensure that it is complete and correct. If you have cited papers that have been retracted, please include the rationale for doing so in the manuscript text, or remove these references and replace them with relevant current references. Any changes to the reference list should be mentioned in the rebuttal letter that accompanies your revised manuscript. If you need to cite a retracted article, indicate the article’s retracted status in the References list and also include a citation and full reference for the retraction notice.

Reviewers' comments:

Reviewer's Responses to Questions

**Comments to the Author**

1. Is the manuscript technically sound, and do the data support the conclusions?

Reviewer #1: Yes

Reviewer #2: Yes

2. Has the statistical analysis been performed appropriately and rigorously? 

Reviewer #1: Yes

Reviewer #2: Yes

3. Have the authors made all data underlying the findings in their manuscript fully available?

Reviewer #1: Yes

Reviewer #2: Yes

4. Is the manuscript presented in an intelligible fashion and written in standard English?

Reviewer #1: Yes

Reviewer #2: Yes

5. Review Comments to the Author

Reviewer #1: PDF of reviews attached (same as below)

Article name: Confusion2Vec 2.0: Enriching ambiguous spoken language representations with subwords

This is a very interesting paper – a marriage between acoustic knowledge and language processing. I think the direction of this research is great, and you are harnessing knowledge from two systems that have been well-researched for a very long time. The paper is well-written, conducting analysis from different angles with adequate reasoning for various design choices. The details of the experiments are also provided, along with the code which makes the study reproducible. Especially, the visualisations are pretty cool, and the essence of what is being captured by the models is very clear. However, the paper needs to focus on a few more aspects like:

• the potential application of such embeddings where acoustically similar words have a similar feature space (a potential application where we can use these embeddings instead of an ASR).

• the resource requirement for these embeddings

• what information are the newly developed embeddings missing

I think the authors have this information already, but they just have not reported it. Below are my comments and revision points organised at the section level.

Abstract: The abstract is well-written and the contribution of the paper is clear. However, include a statistical evidence of the significantly better performance that is claimed eg: an xx% increase)

Introduction:

1. “Although, there have been few attempts in leveraging information present in word lattices and word confusion networks for several tasks” – this sentence undermines the amount of work that has happened with word lattices and confusion networks. Even the references you have mentioned contain numerous citations. Rephrase the sentence clearly stating that the representations using lattices and confusions networks have been successful in multiple tasks, however, they have some limitations.

2. The motivations need to be referenced. For example, sentences like:

• the acoustically ambiguous words tend to have more similar bag-of-character n-grams

• subwords help model under-represented words more efficiently

• subwords enable representations for out-of-vocabulary words

3. Somewhere in the introduction the difference of this study from the initial Confusion2Vec model has to be clearly mentioned.

4. “the main downside with these works is that the word representation estimated by such techniques are task-dependent and are restricted to a particular domain and dataset.” – has this been experimentally verified? If yes, state the reference. If no, this sentence will have to be rephrased. If we have a large text database of a language (which often exists in atleast the well-resourced languages like US English) and a relatively smaller domain-specific text databases, the representations should still be good for the domain-specific task. As for the speech database, dealing with “unseen” words in ASRs is a problem that is more general than specific to this paper’s theme.

Confusion2Vec:

• The section heading should be a bit more descriptive – maybe Confusion2Vec representation framework?

Confusion2Vec 2.0 subword model

• “We believe we have a compelling case for the use of subwords for representing the acoustic similarities (ambiguities) between the words in the language since more similarly sounding words often have highly overlapping subword representations.” – reference for this statement? More clearly, why do you think it’s a good representation?

• “use of subwords should help in efficient encoding of under-represented words in the language.” – reason for this or reference?

• “In the proposed model, each word w is represented as a sum of its constituent n-gram character subwords.” – Replace with - “In the proposed model, for example, … “

• “The n-grams are generated for n=3 up to n=6.” – Why? Why is n=3 and n=6 the maximum and minimum limits for English? Is this based on language analysis, if yes, provide references.

• “The n-grams are generated for n=3 up to n=6.” Is this language-dependent? Would someone working on another language be safe to use n=3 and n=6?

Training Loss and Objective

• Equation (4)’s description is ambiguous. Is the equation that of the binary logistic loss? It is the objective function for subword model’s negative sampling. Please mention that clearly, and add an LHS to this function.

• Are there any other differences between the new implementation and the old Confusion2vec? Other than that, you are using subwords here? Again, any other changes should be clearly mentioned.

Evaluations:

• “useful, meaningful information embedded in the word vector representation” – what is the difference between useful and meaningful in this context? Can it be useful but not meaningful, or can it be meaningful but not useful?

• For all the databases you have used, clearly mention the language of the database, and size of the database. This is essential for someone trying this out in another language.

• W2V – first time usage needs to full form.

• You mention the Word Similarity task – did you use human annotators for this? Or did you just use the results from [20] – in either case that has to be mentioned clearly, including number of people who annotated.

• A description of the results is needed. Did your evaluation show that Confusion2vec 2.0 is better or comparable to existing representations?

• What do the bold numbers in the Appendix Table 5 mean?

Analogy & Similarity Tasks

• Automatic speech recognition – how important is the performance of the ASR to the Confusion2Vec training? Will an ASR with better performance (better WER) be better for the Confusion2Vec training?

• “Also, a minimum frequency threshold of is set and the rarely occurring words are pruned from the vocabulary.” – what is the motivation for this other than reducing the training time/resources? Will Confusion2Vec representation be able to deal with “unseen” words?

• Under the results section – are the analogy and similarity tasks performed on the 353 pairs? Please refer to the relevant section.

• Why do you think Confusion2Vec 2.0 performance is lower compared to Confusion2Vec and FastText for S&S analogy task?

• “Investigating the results for the similarity tasks, we find a significant correlation of …” – how was this correlation calculated? Did you have annotators perform the task for you? Or used the results from past annotations?

Model Concatenation

• “The subword models slightly under-perform in the acoustic analogy task …” This is a very interesting result and contradictory to what we expect. Why do you think this is the case? It feels that in these concatenations, the impact of fastText is dominant than Confusion2Vec.

• This is a general comment for all the training you have mentioned in this paper. To allow resproducibility of your results, and to allow other researchers to judge whether the resources they have are sufficient to undergo your experiments – please provide details of your computational resources and the training time needed. This maybe a separate section, or even included in Appendices.

Embedding Visualization

• Give details of the packages used for the visualisation.

• The example about “prinz” is interesting – but is this a one-off example? Are there other occurrences of words that are clustered together due to their acoustic similarity? Also, was prinz part of the training set?

• It would be interesting to see a similar visualisation of Confusion2Vec 1.0 and the concatenated model too so that a comparison can be drawn with Confusino2Vec 2.0. I had a look at the Confusion2Vec 1.0 paper, but, as the same word list is not used, a direct comparison is not possible.

• The visualization is interesting and gives a clear picture (literally) of what the models are doing. We can see that the Confusion2Vec 2.0 is clearly modelling human perception. But it feels like it is modelling human perception of individual words in isolation without the context. That would describe why it has such a close feature space for “prints” and “prince”. But then, is that good? Do we not want our NLP applications to be able to differentiate these two words rather than consider them as similar? I think this also explains the high correlation you have got in the acoustic similarity tasks – basically where humans are finding individual words acoustically similar, Confusion2Vec 2.0 is also finding the same, and not otherwise. This needs to be addressed in your discussion:

Why is the Confusion2Vec in its embeddings training not capturing the context information? Or rather what can we do to make it capture context information AND acoustic similarity? Maybe the concatenated model is the solution for this. We can know this only by having a look at the visualisation.

In what application would you want acoustically similar words to have a similar feature space? I understand it is good for cases like a noisy ASR output or mispronounced words.

Finally, what impact did the sub-word model bring here that a word-based model could not?

Spoken Language Intent Detection

• In the Database section – what does “samples” mean? Sentences?

• “Among the different versions of the proposed subword based Confusion2vec, we find that the concatenated versions are slightly better.” – It does not look like they are “slightly” better, it looks like they are clearly better. Again, I think the visualisation of the concatenated models in the visualisation section is essential.

• Please provide some examples of the Intent detection task – sentences, along with human-annotated intent and ASR identified intent.

• “This confirms our initial hypothesis that the subword encoding is better able to represent the acoustic ambiguities in the human language.” – are we sure that this experiment is proving that? The statement is ambiguous because it feels like the model is able to differentiate the ambiguous words – rather from the visualisation we see that it is clustering the ambiguous words together. Hence, this claim has to be made unambiguous. Also, the results are good for this particular task, or in tasks were its okay to have similar representation for ambiguous words. What about applications where a differentiation is needed?

• “These results prove that the subword-Confusion2vec models can eliminate the need for re-training natural language understanding and processing algorithms on ASR transcripts for robust performance.” – again too generalized – this is an intent classification task and the experiment only proves the efficacy of the model for this task or similar ones. It should not be generalized.

Conclusion

• A discussion section needs to be added that discusses the impact of the findings of this paper. A few points that can be discussed about are:

o The impact of having context information for the Confusion2Vec embeddings.

o Some applications of Confusion2Vec 2.0 – like what is the use of clustering similar sounding words together for an NLP application – especially without context information.

o The resource requirement for this embeddings development – For some languages it may not be possible to have a large, annotated speech database for ASR modelling. So, it is important to state how much speech and text data is required.

Reviewer #2: The article proposed a Confusion2Vec 2.0 to handle the ambiguities found in natural language using subword modeling units. The article presents the performance over various evaluations tasks including word analogy and word similarity tasks, which deal with acoustic, syntactic, and semantic ambiguities. The empirical evaluations presented in the article are thorough and have significant improvements over the existing methods.

Overall, the research article is mostly clear when it comes to related literature, methodology, and result analysis. The language is simple enough to read and understand. However, there are few flaws and questions throughout this article that authors should consider and clarify, and are mentioned below:

1. There are many state-of-the-art end-to-end ASR models exist today, why the traditional HMM-DNN based pipeline has been used?

2. Have you considered other modeling configurations other than inter and intra-confusion?

3. The table captions should include the metrics of the results. For Eg. From table #1 and #2, it is tough to figure out what kind of results are presented.

4. There are no red lines and ellipses in Figure 2. I believe It should be orange.

5. There are many grammatical errors in the article. The examples can be found in lines #279 and #313, where “are” should be “is”. Further, line #312 “is” should be “are”.

6. There is some unnecessary hyphenation. Eg. Common-place (commonplace), hyper-parameters (hyperparameters).

7. Acronyms in the reference section should be in uppercase letters. Eg. Bert (BERT), Rnn (RNN), Asr (ASR), Glove(GloVe), and so on.

8. Missing full stop on line #194.

9. The English article should be used wherever possible.

6. PLOS authors have the option to publish the peer review history of their article (what does this mean?). If published, this will include your full peer review and any attached files.

Reviewer #1: No

Reviewer #2: No

---

## [Author Response · Author response to Decision Letter 0]

11 Feb 2022

Please see attached Response_to_reviewers.pdf for detailed response.

---

## [Editor Report · Decision Letter 1]

14 Feb 2022

Confusion2Vec 2.0: Enriching ambiguous spoken language representations with subwords

PONE-D-21-33107R1

Dear Dr. Gurunath Shivakumar,

We’re pleased to inform you that your manuscript has been judged scientifically suitable for publication and will be formally accepted for publication once it meets all outstanding technical requirements.

Kind regards,

Seyed Reza Shahamiri

Academic Editor

PLOS ONE

---

## [Editor Report · Acceptance letter]

24 Feb 2022

PONE-D-21-33107R1 

Confusion2Vec 2.0: Enriching ambiguous spoken language representations with subwords 

Dear Dr. Gurunath Shivakumar:

I'm pleased to inform you that your manuscript has been deemed suitable for publication in PLOS ONE. Congratulations! Your manuscript is now with our production department. 

Kind regards, 

on behalf of

Dr. Seyed Reza Shahamiri 

Academic Editor

PLOS ONE